# A Data-Driven DAE-CNN-BiLSTM-Attention Prediction Model for the State of Health of Lithium-ion Batteries

1st Can Zhang
School of Electrical Engineering
Southwest Jiaotong University
Chengdu, China
zhangcan@my.swjtu.edu.cn

2nd Yuanjiang Hu
School of Electrical Engineering
Southwest Jiaotong University
Chengdu, China
360832800@qq.com

3rd Deqing Huang
School of Electrical Engineering
Southwest Jiaotong University
Chengdu, China
elehd@home.swjtu.edu.cn

4th Jiaxin Fang
School of Electrical Engineering
Southwest Jiaotong University
Chengdu, China
2022210644@my.swjtu.edu.cn

5th Na Qin
School of Electrical Engineering
Southwest Jiaotong University
Chengdu, China
qinna@swjtu.edu.cn

*Abstract*—Accurately predicting the health state of lithium-ion batteries is crucial for their safety, reliability, and longevity. Predicting the State of Health (SOH) using health indicators is a proven effective method. However, real-world battery charge-discharge data is often noisy, particularly during capacity regeneration. To achieve accurate health state predictions, we extracted over ten health indicators and designed a hybrid model: DAE-CNN-BiLSTM-Attention. This model integrates the strengths of Convolutional Neural Networks (CNN) for local feature extraction, Bidirectional Long Short-Term Memory networks (BiLSTM) for temporal dependency learning, the Attention mechanism for effective weight assignment, and Denoising Autoencoders (DAE) for restoring original data, enabling the network to better adapt to complex real-world environments. The adaptability and stability of the proposed model were validated using two public datasets: NASA and CALCE. Compared to other existing methods, the proposed model demonstrated superior performance, achieving mean absolute error (MAE) and root mean square error (RMSE) of 0.0154 and 0.0191, respectively.

*Index Terms*—State of Health(SOH), Lithium-ion Battery(LiB), feature extraction, convolution neural network (CNN), long short-term memory (LSTM)

## I. Introduction

To address environmental challenges and the fossil energy crisis, there is an urgent and vigorous development of clean energy sources such as wind, hydro, and nuclear power. Consequently, the issue of energy storage and utilization has become particularly critical. Lithium batteries, compared to other types such as lead-acid and nickel-cadmium batteries, offer higher energy density, lower self-discharge rates, and longer charge-discharge lifespans. These advantages have led to their widespread application in various fields, including portable electronic devices, electric vehicles, and energy storage systems [1].However, over time and with usage, batteries inevitably experience aging. This results in increased internal resistance, reduced usable capacity, and degraded performance, which can lead to battery leakage, localized short circuits, and potential safety hazards such as device malfunctions, shutdowns, or even overheating and explosions. Consequently, in critical applications, batteries are often replaced periodically to ensure safety, which inevitably leads to resource wastage. Battery Management Systems (BMS) play a crucial role in ensuring the safe, reliable, and efficient operation of batteries, with the State of Health (SOH) being a core concern. Accurately predicting the SOH is vital for assessing battery aging, conserving resources, and ensuring the safe operation of batteries [2]- [4].

The State of Health (SOH) of a battery reflects its degree of aging, typically expressed as a percentage that decreases as the battery ages. It is manifested through parameters such as reduced total usable capacity and increased resistance. The higher the SOH value, the healthier the battery. SOH estimation can be achieved by monitoring parameters such as the battery's voltage, current, temperature, and other factors. Commonly, battery health is represented by the ratio of the maximum usable capacity to the nominal capacity5.This paper also adopts this definition, with the SOH defined as shown in Eq (1).

$$SOH = \frac{C_{\max}}{C_{\mathrm{nom}}} \times 100\% \qquad (1)$$

where $C_{\max}$ and $C_{\mathrm{nom}}$ are the maximum usable capacity and the nominal capacity of the battery, respectively. In most applications, a decline in the battery's maximum usable capacity to 70% of its initial capacity is generally considered the failure threshold [6].

Due to the complex operating environments of batteries, such as temperature variations and the internal chemical reactions within the battery, which introduce uncertainties, the highly nonlinear and time-varying characteristics of batteries make accurately predicting SOH a challenging research problem [7]. Currently, many methods exist to accurately predict battery SOH, which can be mainly classified into model-based and data-driven approaches. Model-based methods predict battery SOH by acquiring battery model parameters. These methods analyze and utilize the physical and internal chemical characteristics of the battery to establish equivalent circuit models [8]- [9] or electrochemical models [10]. Typically, state observers are used to describe the degradation mechanisms between battery cycles [11], such as Kalman filters [12]- [13] and particle filters [15]. Although electrochemical models have relatively high accuracy, they rely on precise electrochemical impedance spectroscopy. On the other hand, equivalent circuit models are less satisfactory because they fail to capture the aging characteristics of the battery. Model-based approaches often involve ideal or empirical models that do not account for internal chemical reactions and aging mechanisms, making accuracy increasingly difficult to maintain over time [1]. Additionally, the physical and chemical parameter models of batteries are very complex, which imposes severe limitations due to measurement difficulties, robustness, dynamic accuracy, and poor adaptability.

In contrast to model-based methods, data-driven approaches do not require consideration of these parameters. Instead, they directly extract and analyze historical charge and discharge data from the battery, performing deep data mining to identify the relationship between the extracted features and the battery SOH through machine learning or deep learning techniques. Examples include Backpropagation (BP) neural networks [16], Support Vector Machines (SVM) [17], Relevance Vector Machines (RVM) [18], and Bayesian networks [19]. However, considering the time dependency of battery degradation data, recurrent neural networks (RNNs) have shown superior predictive performance. Literature [11] has already suggested using RNNs for battery SOH prediction. Long Short-Term Memory (LSTM) networks [20], as an upgraded version of RNNs, prevent issues like gradient explosion and perform exceptionally well in sequence prediction. To connect the degradation data over time, some studies have used bidirectional LSTM networks [21]. To overcome the limitations of single networks, more research has adopted hybrid network approaches. For example, [22] used LSTM combined with Empirical Mode Decomposition for predicting Remaining Useful Life (RUL) of batteries. Paper [23] combined Gated Recurrent Units (GRU) with Convolutional Neural Networks (CNN) (CNN-GRU) to predict SOH of lithium-ion batteries. Study [24] employed a hybrid network composed of CNN and LSTM (CNN-LSTM) for SOH estimation and RUL prediction.

Besides using different algorithms for SOH estimation, many studies have extracted various Health Indicators (HIs) to improve SOH estimation results for lithium-ion batteries. Indicators include Constant Current-Constant Voltage (CC-CV) protocols [25], Open Circuit Voltage (OCV) [26], Incremental Capacity (IC) curve peaks [27], cycle numbers [28], differential capacity [29], and differential voltage [30], which describe battery degradation. Reference [31] selected external characteristic parameters such as current, voltage, and temperature as HIs and used the Pearson correlation coefficient to calculate HIs that are highly correlated with the SOH degradation process of lithium-ion batteries.

Despite achieving good prediction results, most methods assume that hidden layer features obtained from raw input data by neural network (NN) training have equal weight for each dimension. However, each feature dimension has different impacts on SOH. Ignoring this fact can limit prediction accuracy [11]. Attention mechanisms, including channel attention (i.e., dimension attention), multi-head attention, spatial attention, and temporal attention [32], focus on important information. Attention mechanisms allow network models to selectively focus on specific information that is more valuable for the current task, thereby improving model performance. Moreover, real-world data are often noisy, especially during the battery charge and discharge capacity regeneration process [33].

In summary, this paper proposes a novel hybrid network model. Initially, the data is denoised, followed by the extraction of local features using two convolutional layers. Long-term dependencies within the sequence are captured through a bidirectional LSTM, while a temporal attention mechanism focuses on each timestep in the sequence, assigning a weight to each timestep to emphasize important points and improve the handling of sequential data. The main contributions of this paper are as follows:

1. The accuracy and feasibility of the DAE-CNN-BiLSTM-Attention model for SOH prediction were validated on two commonly used public datasets, NASA and CALCE.

2. Ten health indicators related to battery aging, including time, temperature, voltage, current, and internal resistance, were extracted. To avoid the interference of multiple factors, the top five most relevant health indicators for each battery were selected as network input features.

3. The model considered the phenomenon of battery capacity regeneration and the impact of real-world noise by incorporating a denoising step in the code, enhancing the model's robustness.

The remainder of this paper is organized as follows: Section 2 describes the methods used, including feature extraction and the proposed network model. Section 3 validates the model's effectiveness with actual battery data, presenting experimental results and analysis. Section 4 provides the conclusions of this study.

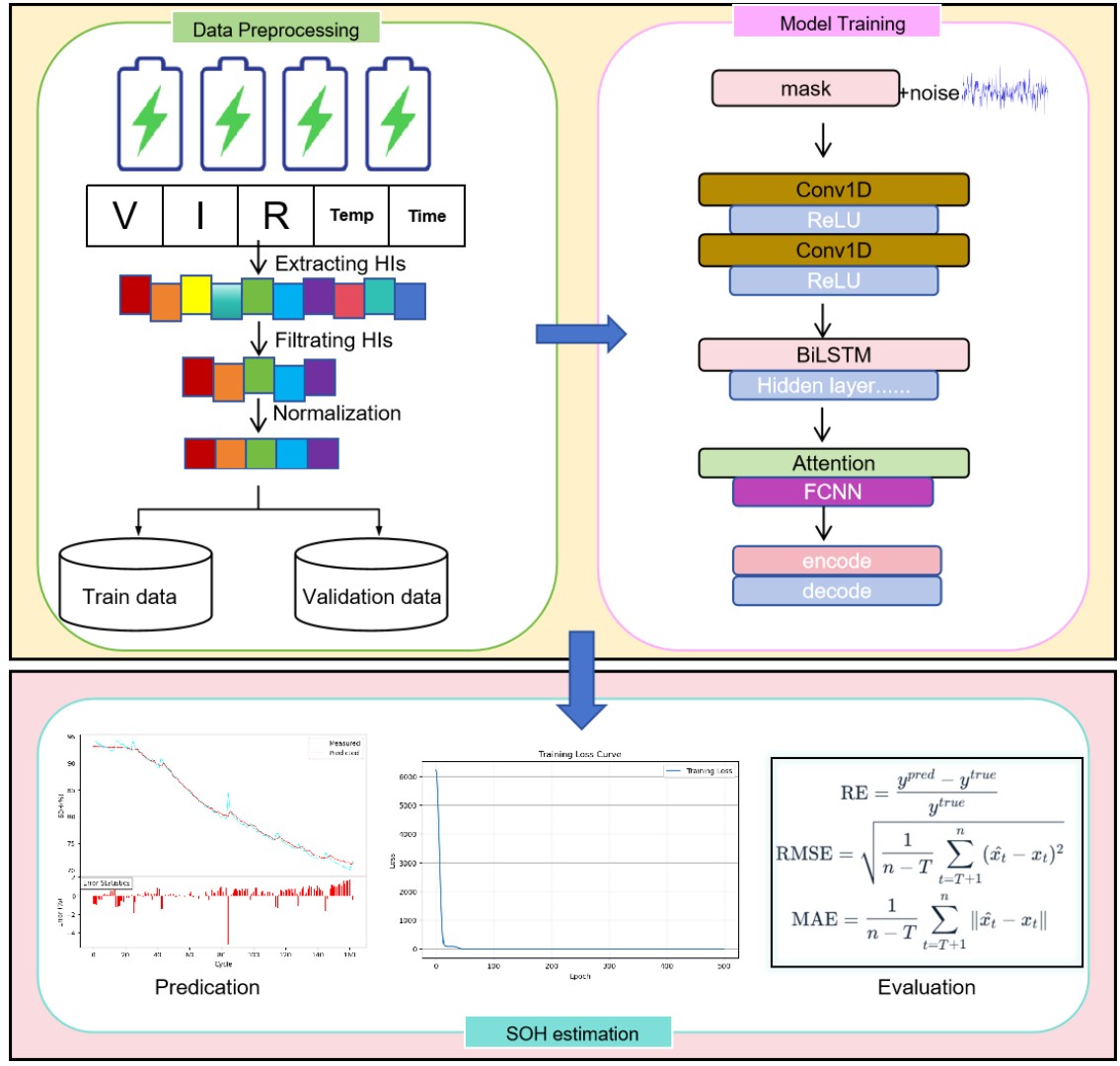

Fig. 1: Overall framwork of the proposed battery SOH estimation model.

## II. Methdology

### A. Feature extraction

The data-driven health indicators are derived from the datasets. All health indicators are extracted from the NASA and CALCE datasets. The extracted simple indicators and their aging performance are shown in Table I.

These health indicators are multidimensional features, each with varying degrees of correlation to the SOH. Including all health indicators in the output could introduce noise from less relevant features, thereby reducing prediction accuracy. Therefore, we eliminate low-correlation indicators and select high-correlation indicators for input into the network. In this study, we use Pearson correlation coefficient analysis to filter the indicators, ultimately selecting the top five features for network input. The Pearson correlation coefficient is commonly used to analyze the

correlation between health indicators and SOH [34], and its calculation principle is shown in Eq (2).

$$r = \frac{\sum_{i=1}^{n}(x_i - \bar{x})(y_i - \bar{y})}{\sqrt{\sum_{i=1}^{n}(x_i - \bar{x})^2}\sqrt{\sum_{i=1}^{n}(y_i - \bar{y})^2}} \quad (2)$$

where $x_i$ and $y_i$ represent the values of the data points, with $\bar{x}$ and $\bar{y}$ denoting their respective mean values, and $n$ being the total number of data points.The Pearson correlation coefficient $r$ quantifies the degree of linear relationship between two variables and ranges from -1 to 1. The closer the absolute value is to 1, the stronger the correlation. A value closer to 1 indicates a strong positive correlation, while a value closer to -1 indicates a strong negative correlation.An absolute value of $r$ close to 1 signifies a strong correlation. Specifically, an $r$ value approaching 1 indicates a strong positive correlation, while an $r$ value approaching -1 suggests a strong negative correlation.

TABLE I: Details of Health Indicators (HIs)

| Abbreviation | Explanation | Aging Behavior |
|---|---|---|
| CCT | Constant current charging time | Shortens as battery ages due to increased internal resistance causing faster voltage rise |
| CVT | Constant voltage charging time | Lengthens as battery ages, with reduced current acceptance near full charge, lowering charging efficiency |
| DT | Discharge time | Shortens as battery ages, with increased internal resistance causing faster voltage drop |
| TT | Time to reach maximum temperature | Shortens as battery ages, with increased internal resistance generating more heat, causing faster temperature rise |
| R | Internal resistance | Increases as battery ages |
| CMT | Time for constant voltage charging current to drop to 1.5A | Shortens as battery ages, with decreased capacity and increased internal resistance causing faster current drop |
| CVI mean | Mean constant voltage charging current | Decreases as battery ages, with increased internal resistance and current dropping to a lower level until fully charged |
| CVI std | Standard deviation of constant voltage charging current | Increases as battery ages, with greater fluctuation |
| CCV mean | Mean constant current charging voltage | Decreases due to increased voltage drop from higher internal resistance |
| CCV std | Standard deviation of constant current charging voltage | Increases as battery ages, with greater fluctuation |
| CDV mean | Mean constant current discharging voltage | Decreases due to increased voltage drop from higher internal resistance |
| CDV std | Standard deviation of constant current discharging voltage | Increases as battery ages, with greater fluctuation |

## B. DAE-CNN-BiLSTM-Attention model

The raw input data is often noisy, especially during charge and discharge cycles. In most methods, the raw data is fed directly into the neural network without any denoising, which significantly impacts the prediction accuracy. To ensure stability and robustness, it is essential to denoise the input data before feeding it into the deep neural network. Our approach uses a denoising autoencoder (DAE), an unsupervised learning method that reconstructs the input data from its low-dimensional representation while preserving as much information as

possible [35].

Attention mechanisms have been widely applied in various deep learning tasks in recent years [36]. In this paper, we calculate attention scores using an attention mechanism, convert them into weights with the softmax function, and then apply these weights to the outputs of the LSTM to obtain context vectors. This approach is simple to implement, computationally efficient, and well-suited for handling time-series data, highlighting important temporal information within the network.

Fig. 1. illustrates the framework of the DAE-CNN-BiLSTM-Attention model for predicting battery SOH, including denoising functionality and the CNN, BiLSTM, and Attention modules. In part A, health indicators are extracted, and the top 5 features are selected based on their Pearson correlation coefficients. These features are then normalized, with 70% used as training data and 30% as validation data. Gaussian noise is added, followed by two CNN layers to extract local features. The BiLSTM captures long-term dependencies in the sequential data, while the attention mechanism helps the network focus on the most important parts for predicting SOH. An autoencoder is employed to denoise the data by attempting to reconstruct the original data from the noisy input, thereby enhancing the robustness of the network.The model structures are summarized in Table II. The loss function converges to zero, and the model's performance is quantified using RMSE and MAE metrics.

## III. Experiment results and analysis

### A. Datasets

The data from the NASA repository was collected by the NASA Ames Prognostics Center of Excellence (PCoE) on the NASA prognostics testbed [11]. NASA batteries were used to validate the proposed method [35]. This study utilizes batteries B0005, B0006, B0007, and B0008, each undergoing charge and discharge cycles at 24°C with a nominal capacity of 2Ah. During the charging phase, the batteries were charged with a constant current of 1.5A up to 4.2V, followed by constant voltage charging until the current dropped to 20mA. In the discharging phase, the batteries were discharged with a constant current of 2A until the cut-off voltage was reached. This cycle was repeated continuously.Fig. 2 shows the capacity degradation process of the NASA battery dataset.

The CALCE dataset is a battery cycling test dataset from the Center for Advanced Life Cycle Engineering (CALCE) at the University of Maryland. CALCE batteries are widely used in battery state estimation studies and were used to validate the proposed method in [33]. This study uses batteries CS2_35, CS2_36, CS_37, and CS2_38, each undergoing charge and discharge cycles at an ambient temperature of 1°C with a nominal capacity of 1.1Ah. During the charging phase, the batteries were charged with a constant current (CC) of 0.5A until the voltage reached 4.2V, followed by constant voltage (CV)

TABLE II: Structure and parameters of neural networks.

| Model | Structure | Number of Sampling Points |
|---|---|---|
| CNN | noisy input→ $X$ 
 Conv1D(Channel: 64/Kernel: 3/padding: 1)→ReLU→ 
 Conv1D(Channel: 128/Kernel: 3/padding: 1)→ReLU→ | X 
 64 
 128 |
| BiLSTM | Number of bidirectional layers: 1 
 Hidden_size: 100 → Hidden_size * 2 | 128 
 200 |
| Attention | Hidden_size * 2 → 
 Attention_size: 20 → 
 Fc(200→1) | 20 
 200 
 1 |
| Encoder | encoder_fc1: input_size * sequence_length 
 → hidden_size 
 decoder_fc2: → input_size * sequence_length | 100 
 X 
 X |

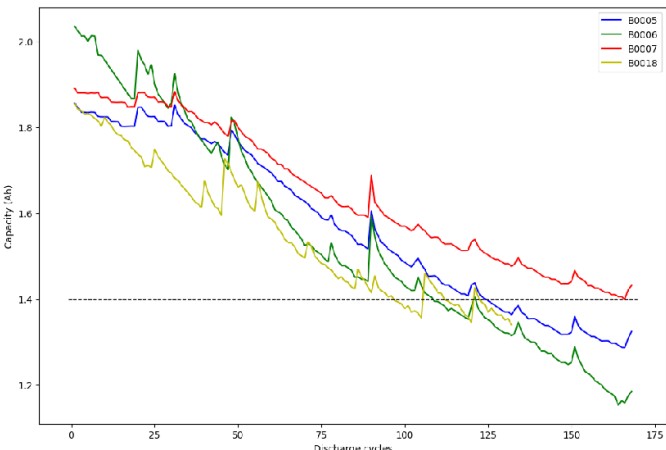

Fig. 2: NASA dataset capacity degration at ambient temperature of 24℃.

charging until the current dropped to 20mA. In the discharging phase, the batteries were discharged with a constant current (CC) of 1A until the voltage dropped to 2.7V.Fig. 3 shows the capacity degradation process of the CALCE battery dataset.

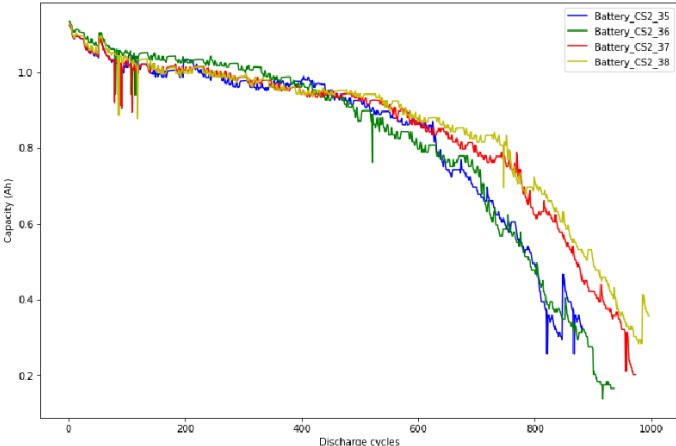

Fig. 3: CALCE dataset capacity degration at ambient temperature of 1℃.

## B. Feature Selection

As shown in table III, the top 5 health indicators are used as inputs to the model. For instance, the inputs selected for B0005 are 'CCT', 'DT', 'TT', 'CMT','CDV mean'. Fig. 4 and Fig. 5 respectively illustrate the correlations between the various health indicators, with red representing positive correlations and blue representing negative correlations.

## C. Overall performance

This study employs three commonly used metrics to quantify the performance of the model in predicting battery health status: Mean Absolute Error (MAE), Root Mean Squared Error (RMSE). The definitions of these metrics are as follows:

$$\text{MAE} = \frac{1}{n-T} \sum_{t=T+1}^{n} \|\hat{x}_t - x_t\| \tag{3}$$

$$\text{RMSE} = \sqrt{\frac{1}{n-T} \sum_{t=T+1}^{n} (\hat{x}_t - x_t)^2} \tag{4}$$

Where $C_n$ represents the length of the sequence, and $C_T$ represents the length of the training sequence samples. MAE (Mean Absolute Error) is the average value of the absolute errors between the predicted and actual values, which measures the average difference between them. RMSE (Root Mean Square Error) is the square root of the average of the squared differences between the predicted and actual values, providing the standard deviation of the errors. RE (Relative Error) is the ratio of the error to the actual value, measuring the proportion of the error in the actual value. The smaller these values are, the better the performance of the model.

We designed several experiments to validate the performance of the proposed model. table IV presents the evaluation results, with the best results highlighted in bold. The DAE-CNN-BiLSTM-Attention model consistently shows the smallest MAE and RMSE, demonstrating superior predictive performance. The best evaluation result for this model was achieved on the CS2_35 battery, with

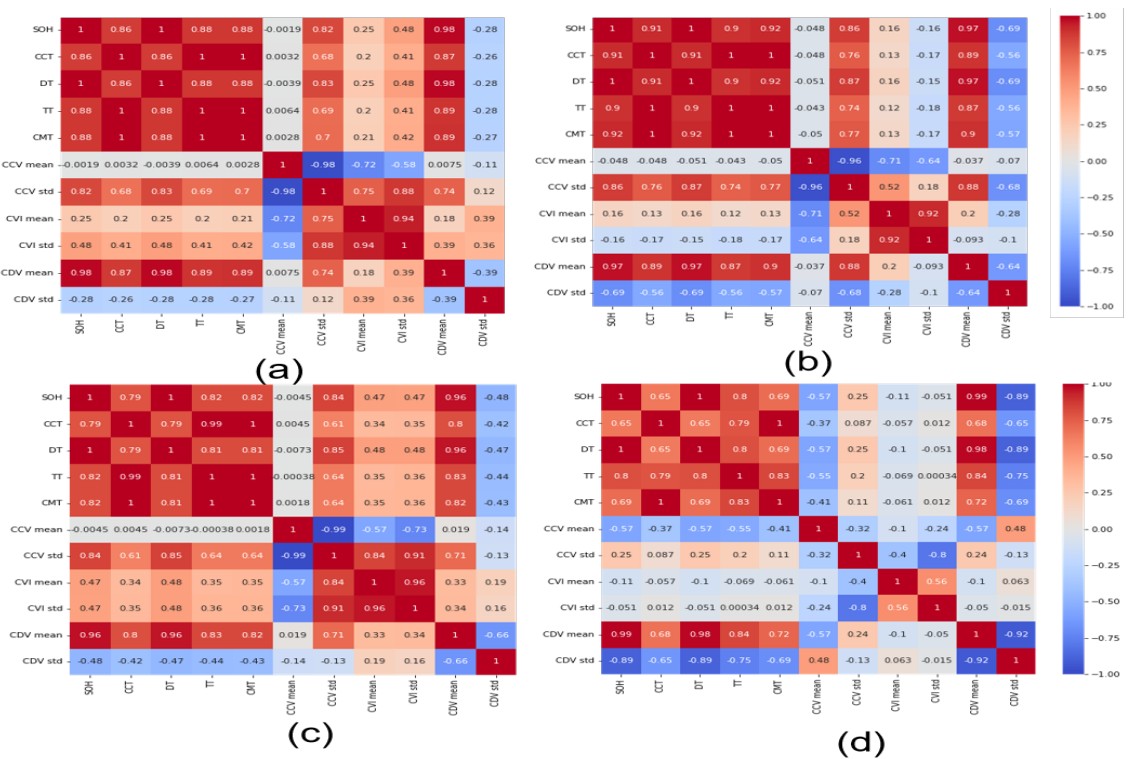

Fig. 4: NASA Pearson Correlation Heatmap:(a)B0005;(b)B0006;(c)B0007;(d)B00018.

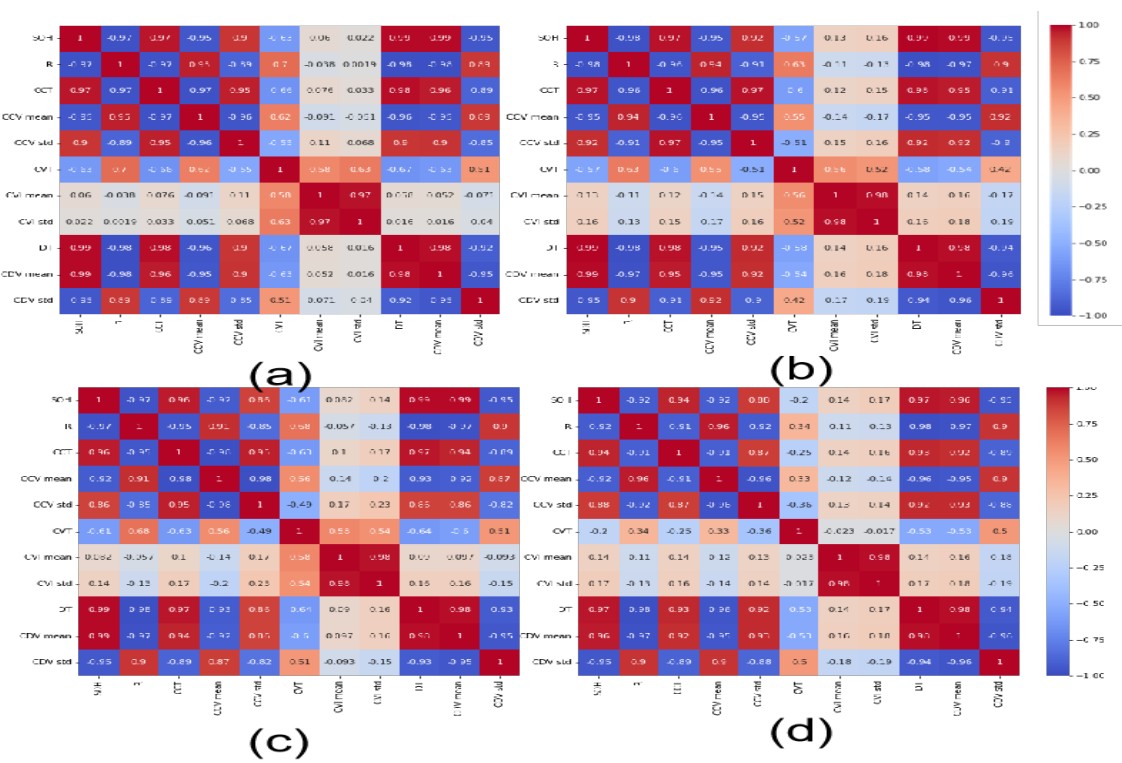

Fig. 5: CALCE Pearson Correlation Heatmap:(a)CS2_35;(b)CS2_36;(c)CS2_37;(d)CS2_38.

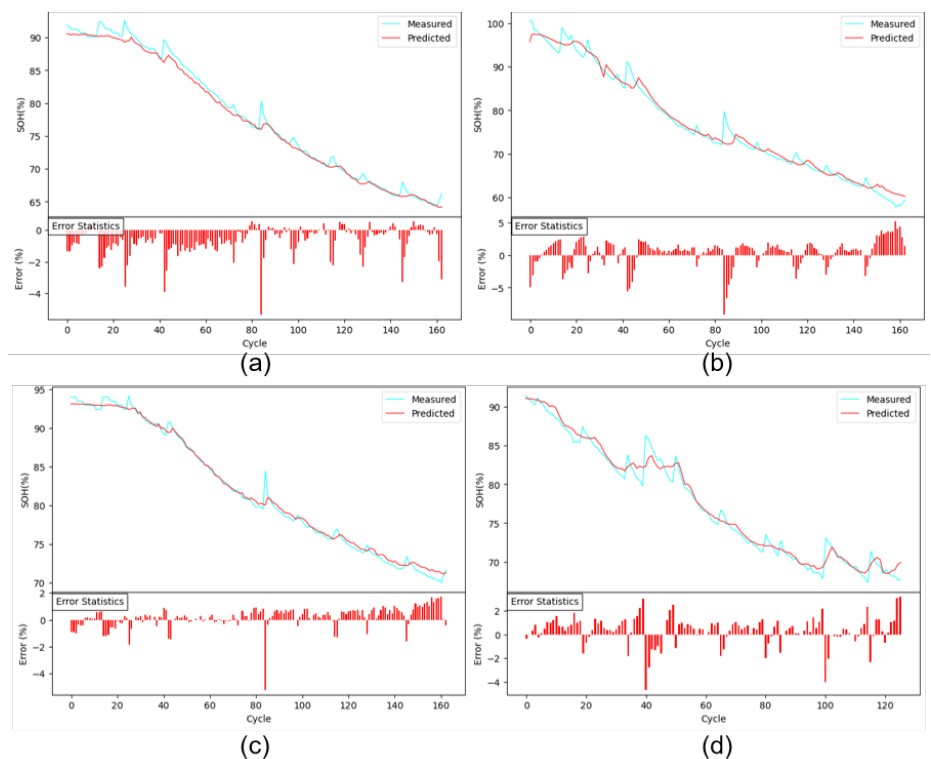

Fig. 6: NASA capacity estimation results and errors:(a)B0005;(b)B0006;(c)B0007;(d)B00018.

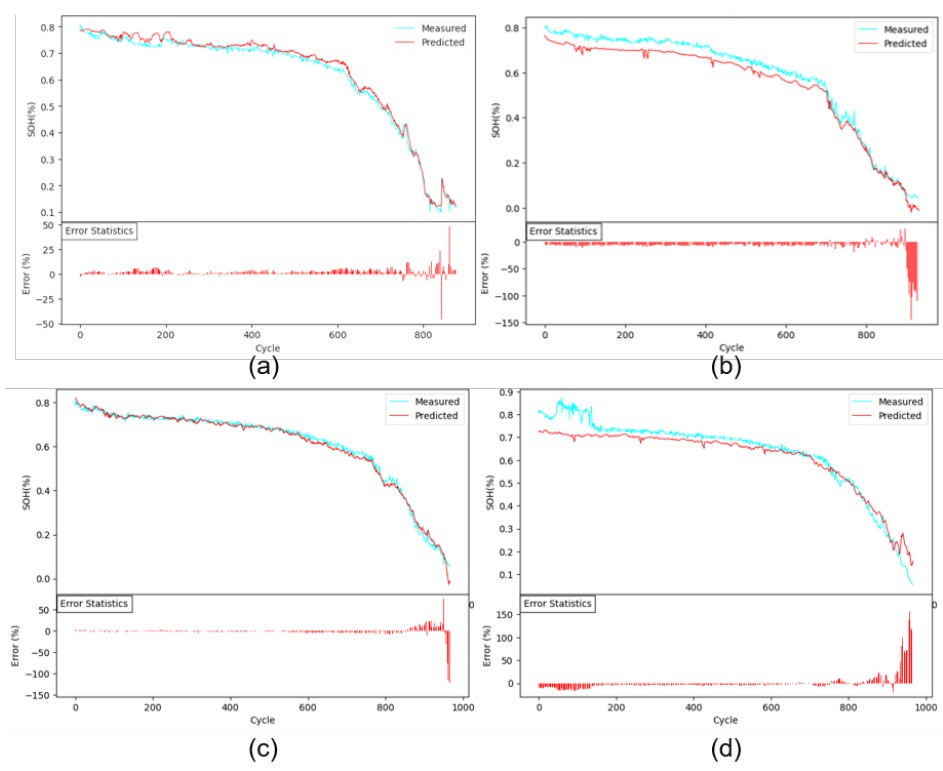

Fig. 7: CALCE capacity estimation results and errors:(a)CS2_35;(b)CS2_36;(c)CS2_37;(d)CS2_38.

TABLE III: Correlation Coefficients of Health Indicators for Different Batteries.

| B0005 | B0006 | B0007 | B0018 |
|---|---|---|---|
| CCT 0.862586 | CCT 0.908604 | CCT 0.786684 | CCT 0.648654 |
| DT 0.999947 | DT 0.999915 | DT 0.999725 | DT 0.999773 |
| TT 0.877558 | TT 0.897890 | TT 0.816621 | TT 0.804534 |
| CMT 0.881882 | CMT 0.919521 | CMT 0.815183 | CMT 0.694932 |
| CCV mean -0.001942 | CCV mean -0.047989 | CCV mean -0.004465 | CCV mean -0.570699 |
| CCV std 0.822106 | CCV std 0.862932 | CCV std 0.839493 | CCV std 0.251537 |
| CVI mean 0.245934 | CVI mean 0.160218 | CVI mean 0.471227 | CVI mean -0.106422 |
| CVI std 0.475350 | CVI std -0.156089 | CVI std 0.468511 | CVI std -0.051191 |
| CDV mean 0.982357 | CDV mean 0.965189 | CDV mean 0.961071 | CDV mean 0.985401 |
| CDV std -0.283559 | CDV std -0.694572 | CDV std -0.482625 | CDV std -0.892615 |
| CS2_35 | CS2_36 | CS2_37 | CS2_38 |
| R -0.969031 | R -0.975631 | R -0.968516 | R -0.922052 |
| CCT 0.967323 | CCT 0.969335 | CCT 0.955910 | CCT 0.942224 |
| CCV mean -0.952510 | CCV mean -0.951288 | CCV mean -0.922861 | CCV mean -0.921456 |
| CCV std 0.897771 | CCV std 0.917804 | CCV std 0.855091 | CCV std 0.875672 |
| CVT -0.626522 | CVT -0.565320 | CVT -0.612713 | CVT -0.197397 |
| CVI mean 0.060142 | CVI mean 0.133853 | CVI mean 0.081966 | CVI mean 0.142492 |
| CVI std 0.022159 | CVI std 0.156692 | CVI std 0.144904 | CVI std 0.173008 |
| DT 0.991876 | DT 0.994180 | DT 0.991499 | DT 0.967231 |
| CDV mean 0.990909 | CDV mean 0.989263 | CDV mean 0.988342 | CDV mean 0.955991 |
| CDV std -0.947330 | CDV std -0.949600 | CDV std -0.945006 | CDV std -0.952266 |

TABLE IV: MAEs and RMSEs of SOH estimation on the NASA and CALCE datasets.

| Datasets | Metrics | LSTM | At-LSTM | CNN-BiLSTM | CNN-BiLSTM-At | DAE-CNN-BiLSTM-At |
|---|---|---|---|---|---|---|
| B0005 | MAE | 1.0882 | 1.0880 | 0.8882 | 0.5521 | **0.5075** |
|       | RMSE | 1.5428 | 1.3567 | 1.3393 | 0.7334 | **0.7064** |
| B0006 | MAE | 1.6684 | 1.2518 | 1.1133 | 1.2459 | **0.8462** |
|       | RMSE | 2.2141 | 1.7345 | 1.6578 | 1.2806 | **1.2405** |
| B0007 | MAE | 1.1695 | 1.1872 | 0.9839 | 0.5992 | **0.4407** |
|       | RMSE | 1.3116 | 1.5503 | 1.4796 | 0.9050 | **0.6337** |
| B0018 | MAE | 1.4277 | 1.2273 | 0.9233 | 0.7266 | **0.7258** |
|       | RMSE | 1.8202 | 1.8140 | 1.2973 | 1.1352 | **0.9738** |
| CS2_35 | MAE | 0.0488 | 0.0470 | 0.0485 | 0.0478 | **0.0154** |
|        | RMSE | 0.0267 | 0.0294 | 0.0228 | 0.0197 | **0.0191** |
| CS2_36 | MAE | 0.0373 | 0.0382 | 0.0341 | 0.0337 | **0.0266** |
|        | RMSE | 0.0391 | 0.0341 | 0.2546 | **0.0230** | 0.0303 |
| CS2_37 | MAE | 0.0315 | 0.0226 | 0.0335 | 0.0371 | **0.0207** |
|        | RMSE | 0.0571 | 0.0262 | 0.0380 | 0.0364 | **0.0335** |
| CS2_38 | MAE | 0.0384 | 0.0358 | 0.0261 | **0.0227** | 0.0286 |
|        | RMSE | 0.0511 | 0.0498 | 0.0522 | 0.0713 | **0.0509** |

a Mean Absolute Error (MAE) of 0.0154 and a Root Mean Square Error (RMSE) of 0.0191. Compared to the model without denoising, the performance improvements in MAE and RMSE were 55.4% and 3.14%, respectively. Additionally, the model is simple, requiring only one minute to complete 500 training iterations, which is significantly faster compared to the 90 minutes and 10 minutes reported in paper [11].

Fig. 6 shows the prediction and error of the NASA battery health state. The predicted results are very close to the actual battery health state values, with all errors within 5% even at peak anomaly points.Fig. 7 shows the prediction and error of the CALCE battery health state.This study utilized 70% of the battery data for training and predicted the entire degradation process. Compared to the NASA dataset, the CALCE dataset has a significantly larger data volume and more anomaly noise, which increases the difficulty of prediction. Although the error has increased somewhat, it still remains close to the actual degradation curve.

## IV. Conclusion

Accurately estimating the State of Health (SOH) of batteries is critical for effective battery management, and es-

tablishing a reliable prediction network is key. We propose a data-driven hybrid neural network for SOH prediction. Initially, we extract over ten features from the batteries and select the top five features based on their absolute Pearson correlation coefficients for input into the network. The Convolutional Neural Network (CNN) first extracts features from the noisy input data, then the Bidirectional Long Short-Term Memory (BiLSTM) network learns the degradation information of the battery, the Attention mechanism focuses on important information, and finally, the autoencoder-decoder restores the noisy data to its original state, enhancing the model's adaptability and stability. The proposed model was validated on different battery datasets and demonstrated lower Mean Absolute Error (MAE) and Root Mean Square Error (RMSE) compared to other models.

## Acknowledgment

The authors would like to thank Teacher Huang for his encouragement and support.

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
