# OpenReview forum: "A Data-Driven DAE-CNN-BiLSTM-Attention Prediction Model for the State of Health of Lithium-ion Batteries"
_IEEE.org/ICIST/2024/Conference — IEEE ICIST 2024 Conference Submission_

### Official Review · Reviewer_fMdt · 2024-08-27
**My Comments for Improvement**

**Rating:** 10
**Confidence:** 4

**Review:**

The paper presents a method utilizing deep learning architectures for the prediction of battery health, which is a critical issue in battery management systems. The paper is well-structured. Here are suggestions for enhancing the paper's academic depth and breadth:

1.	the selection criteria for the "top five health indicators" seem arbitrary. Why use five indicators?

2.	The existing methods for the State of Health of Lithium-ion Batteries should be summarized in the Introduction.

---

### Official Review · Reviewer_H35W · 2024-09-03
**This paper can be considered for publication.**

**Rating:** 6
**Confidence:** 2

**Review:**

The authors in this paper proopose a data-driven DAE-CNN-BiLSTM-attention prediction model for the state of health of
lithium-ion batteries. The reviewer has the following comments for this paper.
1. The presentation quality of this manuscript should be greatly improved.
2. Some future investigation topics should be mentioned in the Conclusion part.

---

### Decision · Program_Chairs · 2024-09-06

Accept (Oral)